# Ecological and Health Risk Assessment of Heavy Metals in Farmland in the South of Zhangbei County, Hebei Province, China

Yanhua Li [2] , Qing Zhu [1], Xuejiao Tang [1], Cuiping Wang [1,*] and Sheng Zhai [2,*]

1 Key Laboratory of Pollution Processes and Environmental Criteria, Ministry of Education, Tianjin Key Laboratory of Environmental Remediation and Pollution Control, College of Environmental Science and Engineering, Nankai University, Tianjin 300071, China
2 School of Geography and Environment, Liaocheng University, Liaocheng 252000, China
* Correspondence: wangcp@nankai.edu.cn (C.W.); zhaisheng@lcu.edu.cn (S.Z.)

**Abstract:** The distribution of heavy metals in the agricultural areas from a cattle-producing area near the Zhangbei Dam grassland, Hebei province, China, was investigated for providing the basis for the control of pollution in a rural farmland. Fifty-three surface soil samples including 28 soils of potato and 25 soils of oats were collected and analyzed for the distribution of Cd, Cu, Pb, Zn, Cr, As, Hg, and Ni. Furthermore, the ecological risk of soils contaminated with heavy metals was evaluated by employing the single factor index, the Nemerow comprehensive pollution index and geo-accumulation index, and potential ecological hazard index. The results showed that the contents of the eight heavy metals in soil planted with potato were below the risk screening values. The single factor index in soils planted with hulless oats showed that the concentrations of Cd, Hg, and Ni surpassed the Chinese screening limits by 8%, 4%, and 8%, respectively. According to the geo-accumulation index, 4% of samples reached the level of medium ecological risk for Cd. According to the results of ecological risk assessment, the studied soils generally showed a mild pollution degree. According to the health risks assessment results, the carcinogenic risks should be considered non-negligible. Arsenic is the dominant carcinogenic pollutant for human beings in the county. The main sources of pollution are mining and application of fertilizers.

**Keywords:** heavy metal; risk assessment; agricultural soil; dam grassland

## 1. Introduction

Health risks for human beings arising from heavy metals have been widely studied [1]. Farmland plays an important role in agricultural production and human survival, so the potential contamination of farmland deserves special attention [2]. In recent years, farmland quality has been compromised by heavy metals pollution originating from industry and unreasonable use of agricultural chemicals [3–6]. Heavy metals are non-biodegradable and toxic in soils [7], affecting the soil physical and chemical properties and soil quality [8]. The national soil pollution survey Bulletin issued by China in 2014 showed that 19.4% of cultivated soil was mainly polluted with heavy metals [9]. The most serious pollutant was Cd, followed by Hg [10]. These elements can trigger ecological risks [11–14]. Heavy metals absorbed by plants in the farmland enter the human body through the food chain, posing health risks to humans [15–19]; especially, Cd and Pb, even at low contents, are very poisonous [20]. An increased risk of cancer is associated with chronic exposure to heavy metals [21]. Exposure to heavy metals threatens human health through oxidative stress and other mechanisms [22]. For example, chronic exposure to Cd is harmful to kidney functions, long-term Pb contact can affect the reproductive system [23], and Zn can cause adverse effects on cholesterol balance and fertility [24]. Hence, it is of great importance to conduct the risk evaluation of heavy metals in the soils. This evaluation is useful to establish the different degrees of pollution and to suggest suitable protection policies for the environment [25,26].

Most studies focused on heavy metals distribution in agricultural soil near the industry or mining areas [27] or in areas irrigated with sewage. For example, a significant pollution of As, Pb, and Hg which existed in Suxian was associated with extensive polymetallic mining [27]; Wang et al. [28] found that soils irrigated with wastewater were contaminated by Cd and Pb in Wuqing, Tianjin, China. However, the distribution of heavy metals in the agricultural areas dominated by a cattle-producing area near Bashang farmland has been insufficiently studied.

The Zhangbei county develops animal husbandry and is a main green crops production base for Beijing-Tianjin-Hebei region and also for the whole of China. The random discharge of livestock manure and misuse of fertilizers have gradually caused pollution in soils. Heavy metals are found to be existing in soils, which pose health threats to the entire ecological system. It is important to study the content and spatial distribution characteristics of heavy metals in the farmland in the agricultural zone, to evaluate the risk of the ecological and human health risk of heavy metals in the soil. Such information provides a basis for the local farmland pollution control and ensures the safety of food.

The potato and hulless oat are the main agricultural products in Zhangbei. In the study, the spatial distribution and ecological risks of heavy metals in potato soils and oat soils were investigated, through combining the methods of pollution index, geo-accumulation index, potential ecological risk index, human health risk, and principal component analysis.

## 2. Materials and Methods

### 2.1. Study Area

Zhangbei county is located in the Bashang plateau area in the northwest of Hebei province, China (114°10′~115°27′ E, 40°57′~41°34′ N). The area is the seat of both animal husbandry and crop farming [29]. The eco-environment is highly vulnerable. The area lies at an average altitude of 1400–1600 m and has a temperate continental monsoon climate. The land use is mainly grassland and crop cultivation. The soil can be mainly classified as a chestnut soil [30]. The soil pH is neutral to alkaline. Agriculture and animal husbandry are predominant in the study area.

### 2.2. Data Sources

Soil samples at a depth of 0–20 cm were collected in September 2021 in the study area (Figure 1), including 28 soil samples planted with potato and 25 soil samples planted with hulless oats. The western region is not the potato or hulless oats main growing area, so the number of collected soil samples is lower than the eastern region. Four or five samples were collected from the site using a plastic shovel. About 100 g from each sample point were mixed and put into a plastic bag. All samples were labeled with sampling location, date and number, and then dried and stored in a pollution-free area.

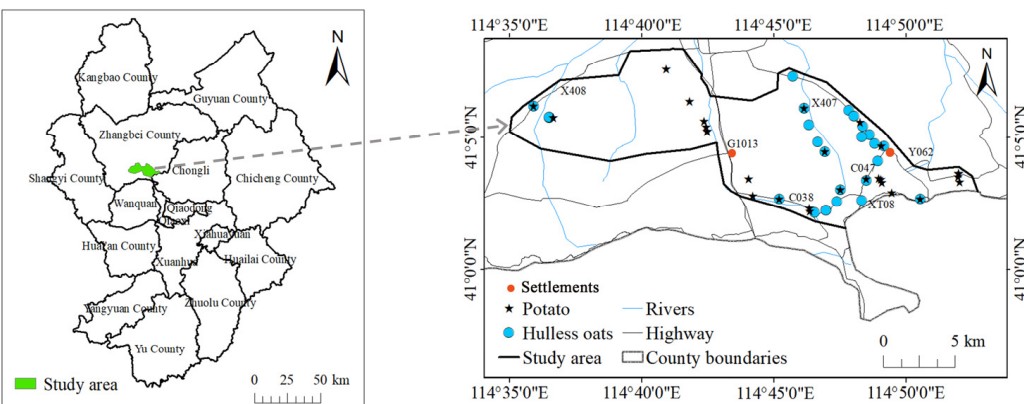

**Figure 1.** Locations of soil sample points in Zhangbei county, Hebei province, China.



According to Chinese agricultural standards of soil testing (NY/T 1121-2006), soil samples were sieved with a 2 mm nylon sieve and 0.15 mm sieve for pH and elements determination, respectively. The pH was determined by the potentiometric method (soil:water = 1:2.5 *w:v*). The soil samples were digested in a microwave oven with a mixture of HNO3 and HF. Cd, Cu, Pb, Zn, Cr, As, Hg, and Ni were determined by inductively coupled plasma mass spectrometry (ICP-MS, Elan DRC-e, PE). Each sample was run in triplicate. The standard soil reference material (GSS-3) was used for quality control and assurance. Standard samples recovery ranged from 90~110%.

*2.3. Pollution Assessment Methods*

2.3.1. Single Factor Index

Single factor index method reflects the pollution status of individual pollutants in soil:

$$P_i = C_i / S_i \tag{1}$$

where $C_i$ means the actual concentrations of heavy metal; $S_i$ means the evaluation standard of pollutants. All evaluation standard values in this paper are referred to the soil pollution screening value of Environmental Quality Standard for Soils (GB15618-2018) (Table S1) [31]. The values of $P_i$ indicate the following [32]: no pollution ($P_i \leq 1$), light pollution ($1 < P_i \leq 2$), mild pollution ($P_i \leq 3$), medium pollution ($3 < P_i \leq 5$), serious pollution ($P_i > 5$).

2.3.2. Nemerow Pollution Index (*NIPI*)

The Nemerow index not only considers the single factor pollution index, but also comprehensively reflects the combined effect of all pollutants:

$$NIPI = \sqrt{\left(P_{max}^2 + P_{ave}^2\right)/2}, \tag{2}$$

where $P_{max}$ and $P_{ave}$ represent the maximum value and average value of single factor indexes, respectively. According to *NIPI*, soil heavy metals pollution can be divided into five levels [33]: safety ($NIPI \leq 0.7$), warning ($0.7 < NIPI \leq 1$), light pollution ($1.0 < NIPI \leq 2.0$), medium pollution ($2.0 < NIPI \leq 3.0$), severe pollution ($NIPI > 3.0$).

2.3.3. Geo-Accumulation Index ($I_{ego}$)

The geo-accumulation index as proposed by Muller [34] is a geochemical criterion to assess the pollution level in sediments or soils. Constant 1.5 was suggested by Muller as an indication of an anthropogenic anomaly. Geo-accumulation not only explains the influences of natural diagenesis on the background values, but considers the influence of anthropogenic activities on the environment [35]:

$$I_{ego} = \log_2\left[C_i \times (K \times S_i)\right], \tag{3}$$

where $C_i$ is the measured concentration of heavy metal; $S_i$ is the background value in the study area; $K$ means the empirical coefficient, which, in general, is 1.5. According to $I_{ego}$, the level of soil heavy metals indicate the following [36]: no pollution ($I_{ego} \leq 0$), low pollution ($0 < I_{ego} \leq 1$), moderate pollution ($1 < I_{ego} \leq 2$), moderate-serious pollution ($2 < I_{ego} \leq 3$), serious pollution ($3 < I_{ego} \leq 4$), serious-heavy pollution ($4 < I_{ego} \leq 5$), heavy pollution ($I_{ego} > 5$).

2.3.4. Ecological Risk Index (RI)

The ecological risk index describes the ecological risk level associated with heavy metals in soil [37]:

$$RI = \sum_{i=1}^{n} E_i = \sum_{i=1}^{n} T_i \times C_i / B_i, \tag{4}$$

where $E_i$ denotes the potential ecological hazard factor of an individual heavy metal, $T_i$ signifies the toxicity response factor, $C_i$ means the measured heavy metal concentration, $B_i$

means the evaluation standard of pollutant. The toxicity response factor for Pb, Cd, Zn, Cu, Ni, Cr, As, Hg are 5, 30, 1,5, 5, 2, 10, 40, respectively. The degree standard of *RI* is presented in Table S2 [38].

### 2.3.5. Human Health Risk Assessment

The study employed the USEPA method, HI, and TCR to assess the non-carcinogenic risk and carcinogenic risk posed by heavy metals to the human body. In the above model, dermal contact, inhalation, and ingestion are assumed as the main exposure routes of heavy metals [39]:

$$CDI_i = (C \times IngR \times EF \times ED \times CF)/BW \times AT \tag{5}$$

$$HQ_i = CDI_i \times RFD_i^{-1}, \tag{6}$$

$$HI = \sum_{i=1}^{n} HQ_i = HQ_{oral} + HQ_{inh} + HQ_{dermal}, \tag{7}$$

where $CDI_i$ is the chronic daily intake of heavy metal (mg/kg/day); $RFD_i$ is the reference dose of heavy metal (mg/kg/day); the meaning of *C*, *IngR*, *EF*, *ED*, *CF*, *BW*, *AT* are shown in Table S3 [40–43]. *HI* > 1 implies that it a non-carcinogenic risk likely exists [44].

Carcinogenic risk (*CR*) denotes the carcinogenic risk caused by heavy metals in the human body:

$$CR_i = CDI_i \times CSF_i, \tag{8}$$

$$TCR = \sum_{i=1}^{n} CDI_i \times CSF_i, \tag{9}$$

where $CSF_i$ is the carcinogenic risk intensity coefficient; $10^{-6} < CR$ and $TCR < 10^{-4}$ indicates that the level of carcinogenic risk is acceptable [45]. The exposure parameter values of health risk assessment have been reported in Table S3. The $RFD_i$ and $CSF_i$ were shown in Table S4 [46–48].

### 2.3.6. Principal Component Analysis (PCA)

Principal component analysis (PCA) can be used to explore relationships among data, in this case, to identify significant sources of heavy metals [49,50]. The data are suitable for PCA when the Kaiser–Meyer–Olkin (KMO) of the dataset surpasses 0.6 [28]. The contents of heavy metals were used as initial values for PCA analysis, and PCs were produced when eigenvalues > 1 [51].

### *2.4. Data Analysis*

Excel 2021 and SPSS 23.0 were used to carry out statistical analysis and PCA. According to characteristics of data, inverse distance weighted was selected for interpolation in ArcGIS 10.5. Origin was applied in the figures.

## 3. Results
### *3.1. Distribution of Heavy Metals in Farmland Soil from the Cattle-Producing Area*
#### 3.1.1. Spatial Distribution of Heavy Metals in Farmland Planted with Potato

Figure 2a1–a8 displayed the spatial distribution of eight heavy metals in soil planted potato. Pb and As had similar patterns of spatial distribution that high concentration mainly distributed in the Dawa village and Bolicai village of western soils, Zhangbei country, China. Cd, Cr, and Hg were greater in northern soil within the survey area, mainly concentrated near X408 County Road. High concentrations of Cd and Hg were observed along roads C047, X407, and Y062 in the eastern area, where urban and rural settlements were distributed. High values of Cr were observed in the northeastern area, distributed in the residential area of Yuanshanzi village, Zhangbei county, China. Cu, Zn, and Ni occurred the high concentrations of distribution patterns, dominantly in the eastern highway crisscross distribution area. In addition, compared to those in the north

of Zhangbei county, Zn in the south of the survey region had a high content. Hence, the high-level blocks of pollution are situated in residential areas and the intersection of highways (Figure 1), suggesting that the increasing of heavy metal in soil was ascribed to substantial human activities [52]. The sources of heavy metal in the agricultural area consisted of fertilization, and the usage of pesticides and fertilization [39].

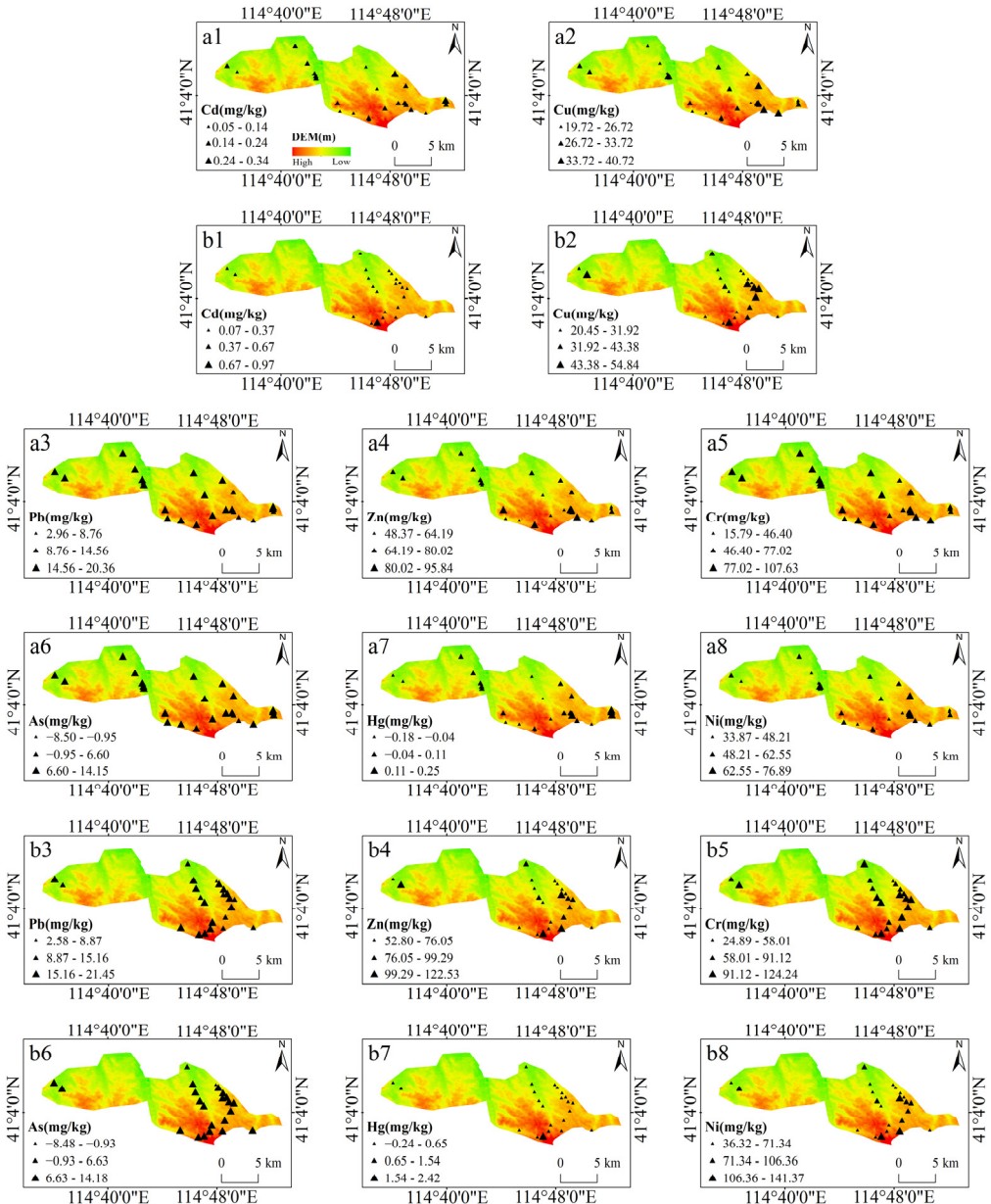

**Figure 2.** Spatial distribution of eight heavy metal contents in soil planted with potato (**a1–a8**) and hulless oats (**b1–b8**) from different villages.

### 3.1.2. Spatial Distribution of Heavy Metals in Farmland Planted with Hulless Oats

There were roughly three spatial distribution types of eight kinds of heavy metals in soil planted with hulless oats (Figure 2b1–b8).

The distributions of Cd and Hg presented an obvious zonal distribution. High concentrations of Cd and Hg focused on the southeast area near C308 compared to other areas. The obvious spatial heterogeneity of Cd and Hg was consistent with the high coefficient of variation of the above Cd and Hg. High Cd and Hg were mainly distributed in Dahonggou village and Xishui village. Cu, Zn, and Ni had similar spatial distribution patterns that

the highest contents dominantly distributed in western areas. Moreover, the high concentrations of Pb, Cr, and As were concentrated in the eastern part of the survey areas. The largest values of Pb were concentrated in the hulless oats in Dahonggou village and Xishui village near C308, China. Cr and As tended to accumulate in the hulless oats in Yuanshanzi village close to the densely populated areas.

*3.2. Concentrations of Heavy Metals in Farmland Soil Dominated by Husbandry*

3.2.1. Concentrations of Heavy Metals in Soil Planted with Potato

Table 1 presents the statistics of heavy metals in soil planted with potato. The pH of the soil ranged from 6.9 to 8.9; 78.6% of the soil samples had a pH greater than 7.5, and the soil was weakly alkaline.

**Table 1.** Concentrations of eight heavy metals in soils planted with potato and hulless oats.

| Element (mg/kg) | Min [1] | | Max [2] | | Median | | Mean | | SD [3] | | CV [4] (%) | | Higher than Risk Screening Value [5] (%) | |
|---|---|---|---|---|---|---|---|---|---|---|---|---|---|---|
| | P [6] | O [7] | P | O | P | O | P | O | P | O | P | O | P | O |
| Cd | 0.05 | 0.07 | 0.34 | 0.97 | 0.18 | 0.17 | 0.18 | 0.20 | 0.06 | 0.17 | 36 | 83 | 0 | 8 |
| Cu | 19.72 | 20.45 | 40.72 | 54.84 | 27.76 | 31.70 | 28.90 | 34.43 | 5.54 | 10.76 | 19 | 31 | 0 | 0 |
| Pb | 2.96 | 2.58 | 20.36 | 21.45 | 16.22 | 15.43 | 15.44 | 14.31 | 3.38 | 4.16 | 22 | 29 | 0 | 0 |
| Zn | 48.37 | 52.81 | 95.84 | 122.53 | 68.68 | 74.82 | 69.39 | 79.01 | 11.63 | 21.82 | 17 | 28 | 0 | 0 |
| Cr | 15.79 | 24.8 | 107.63 | 124.24 | 80.11 | 90.16 | 78.43 | 88.12 | 17.17 | 22.90 | 22 | 26 | 0 | 0 |
| As | −8.50 | −8.48 | 14.15 | 14.18 | 8.43 | 8.58 | 8.37 | 8.37 | 4.08 | 4.29 | 49 | 51 | 0 | 0 |
| Hg | −0.18 | −0.24 | 0.25 | 2.42 | 0.02 | 0.02 | 0.02 | 0.07 | 0.10 | 0.51 | 405 | 716 | 0 | 4 |
| Ni | 33.87 | 36.32 | 76.89 | 141.37 | 51.63 | 56.58 | 53.09 | 72.22 | 15.73 | 31.24 | 20 | 43 | 0 | 8 |
| pH | 6.86 | 6.83 | 8.92 | 8.7 | 8.10 | 8.09 | 8.06 | 7.95 | | | | | | |

[1] Minimum value; [2] Maximum value; [3] The standard deviation; [4] The coefficient of variation; [5] Risk screening value is shown in Table S1; [6] Soil planted with potato; [7] Soil planted with hulless oats.

The mean concentrations of Cd, Cu, Pb, Zn, Cr, As, Hg, and Ni were 0.2, 28.9, 15.4, 69.4, 78.4, 8.4, 0.02, and 53.1, respectively. The average concentrations did not surpass the screening values, showing that there was no pollution risk in the soil. The coefficient of variation (CV) denotes the variation degree between sampling points [40]. It is generally believed that CV ≤ 0.10 was weak variability, 0.10 < CV ≤ 0.90 was medium variability, and CV > 0.90 was strong variability [53]. The greater the coefficient of variation, the more uneven the content of elements in the soil. The results of CV indicated that the Hg had the largest CV, and fell into the category of strong variability, and Cd, Pb, As, Cr, Ni, Cu, and Zn belonged to the medium variability. Such high variation of Hg is probably influenced by human activities, suggesting that there are point sources of pollution [54]. This point, however, requires further investigation

3.2.2. Concentrations of Heavy Metals in Soil Planted with Hulless Oats

The statistics of heavy metals in soil planted with hulless oats is shown in Table 1. The pH of the soil ranged from 6.83 to 8.70. Namely, 72% of the soil samples had a pH greater than 7.5. The soil is weakly alkaline.

Mean concentrations of Cd, Cu, Pb, Zn, Cr, As, Hg, and Ni were 0.2, 34.4, 14.3, 79.0, 88.1, 8.4, 0.07, and 72.2, respectively. The average contents of Cd, Cu, Pb, Zn, Cr, As, Hg, and Ni were lower than the screening value, respectively, indicating that there was no pollution risk in soil planted with hulless oats. However, the Cd, Hg, and Ni at 8%, 4%, and 8% of samples exceeded the screening value, respectively, indicating that Cd, Hg, and Ni were enriched in some samples. Similarly, Cd surpassing the screening value was observed in the farmland of Anxin county of Hebei province [55].

The coefficient of variation of Cu, Cd, Pb, Zn, As, Cr, and Ni fell into the category of medium variability, and CV was highest for Hg (716%), which belonged to the category of strong variability, suggesting that some specific samples have very high concentrations. Such high concentrations may be the result of anthropogenic contamination. About 80% of

anthropogenic sources of Hg result from fuel combustion, waste incineration or exhaust emissions [54]. The sources of heavy metals are different because of different types of land use and surrounding human activities. Thus, the sources of heavy metals in this survey region need to be identified according to a specific manner [44].

### 3.3. Pollution Assessment of Soil Planted with Potato in Bashang Farmland

The highest average value of $P_i$ was for Cd (0.35), followed by Cr (0.33), Ni (0.33), As (0.32), Cu (0.29), Zn (0.24), Pb (0.10), and Hg (0.01). The $P_i$ of eight heavy metals of all samples below level 1 was 100% (Figure 3a), indicating there was no pollution in the study area.

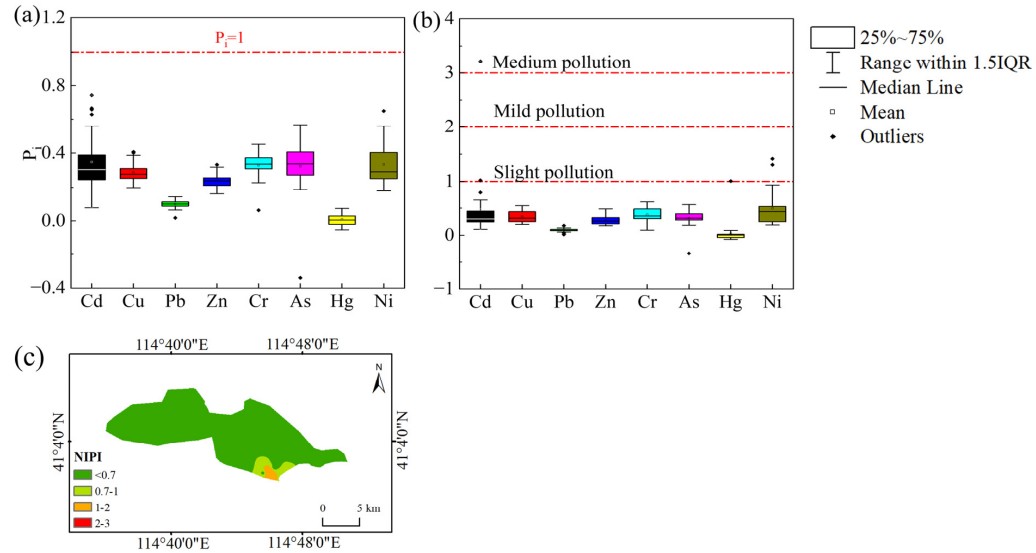

**Figure 3.** $P_i$ of heavy metals in soil planted with potato (**a**) and hulless oats (**b**), respectively; *NIPI* of heavy metals in soil planted with hulless oats (**c**).

The evaluation results of a single factor index illustrated the pollution status of single heavy metals at each sampling location. To understand the overall pollution level of heavy metals at each sampling point, it was necessary to calculate the Nemerow pollution index (*NIPI*). The mean of *NIPI* of soil samples was 0.37, which fell into the safety level.

The $I_{ego}$ values of the eight heavy metals are presented in Figure 4a. The highest mean value of $I_{ego}$ was for Ni (0.24), followed by Cd (0.20), Hg (−0.11), Cu (−0.20), Cr (−0.44), Zn (−0.78), Pb (−1.12), and As (−1.18). For the samples, $I_{ego}$ values of 71.43%, 17.86%, 3.57%, 21.43%, and 85.71% of Cd, Cu, Cr, Hg, and Ni, are at low pollution levels, respectively; 3.57%, 14.29%, and 3.57% of the samples are moderately polluted by Hg and Ni, respectively. In addition, the Hg at 3.57% of the samples are at moderate-serious pollution.

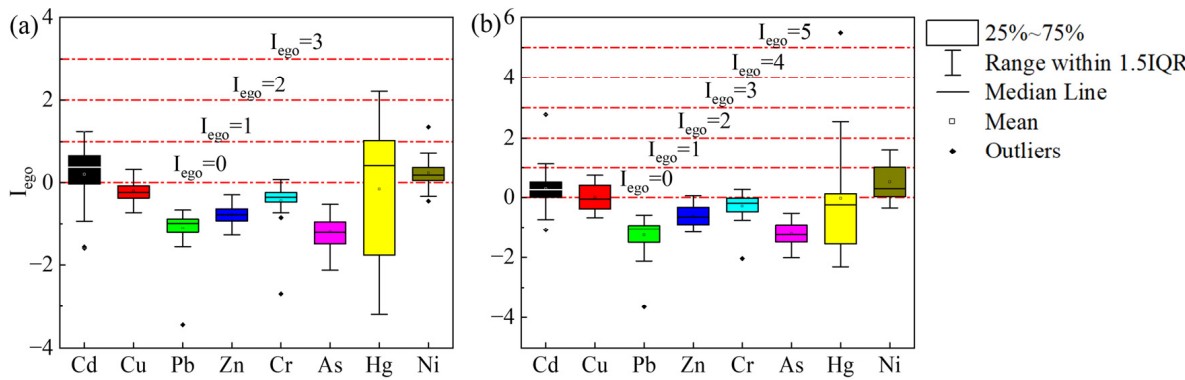

**Figure 4.** $I_{ego}$ of heavy metal in soil planted with potato (**a**) and hulless oats (**b**) from the Bashang farmland.

The $E_i$ of the eight heavy metals were as follows: Cd (10.46) > As (3.23) > Ni (1.67) > Cu (1.45) > Cr (0.66) > Pb (0.49) > Hg (0.33) > Zn (0.24). The $E_i$ of Cd, Cu, Pb, Zn, Cr, As, Hg, and Ni did not exceed 40 (Figure 5a), indicating that the level of ecological risk achieved the light degree. The *RI* of heavy metals in potato soil in the survey area ranged from 0.70 to 32.01, with a mean value of 18.53, indicating that the current content of Cd, Cu, Pb, Zn, Cr, As, Hg, and Ni in soil will not have a potential impact on soil ecological security. Mean contribution rates of the $E_i$ to the *RI* values of eight heavy metals were as follows: Cd (56%) > As (17%) > Ni (9%) > Cu (8%) > Cr (4%) > Pb (3%) > Hg (2%) > Zn (1%).

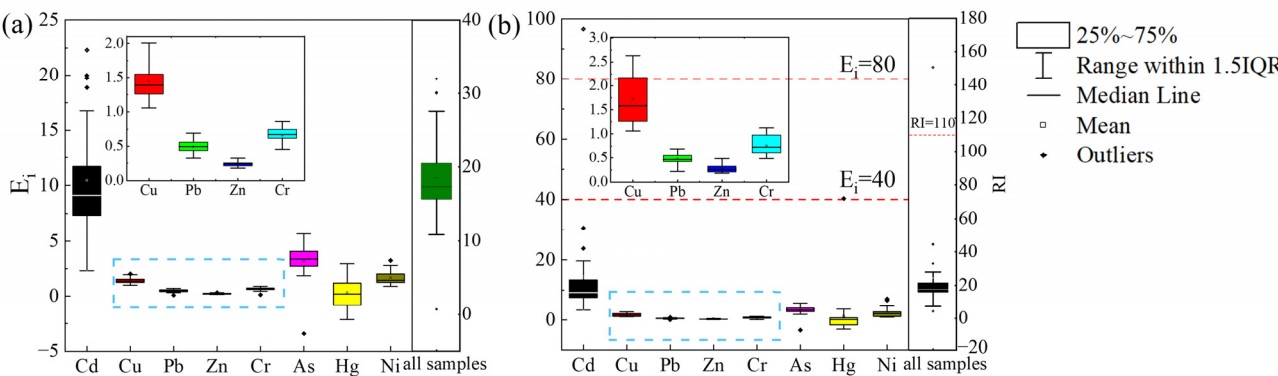

**Figure 5.** $E_i$ of heavy metal in soil planted with potato (**a**) and hulless oats (**b**) in Bashang farmland.

### 3.4. Pollution Assessment of Heavy Metal in the Soil Planted Hulless Oats

Figure 3b shows the $P_i$ value of the eight heavy metals. The mean values of $P_i$ of heavy metals were Ni (0.50) > Cd (0.48) > Cr (0.38) > Cu (0.34) > As (0.32) > Zn (0.28) > Pb (0.09) > Hg (0.03), all of which were less than 1, indicating no pollution in oats soil as a whole. Cd presents a slight degree of pollution in 4% of the samples, while it reaches medium pollution in 4% of the samples; 4% of Hg and 8% Ni, respectively were at the level of slight pollution. The average value of *NIPI* of soil samples was 0.52, which belonged to the safety level. The *NIPI* values ranged from 0.52 to 2.37, suggesting that there are point sources of pollution in some areas (Figure 3c); 12% and 4% of *NIPI* achieved warning and medium level, respectively.

The highest mean value of $I_{ego}$ (Figure 4b) was Ni (0.52), followed by Cd (0.30), Cu (0.01), Hg (−0.01), Cr (−0.28), Zn (−0.62), As (−1.17), and Pb (−1.27). The mean values of $I_{ego}$ of eight heavy metals corresponded to no pollution. For the samples, $I_{ego}$ values of 68%, 44%, 12%, 76%, 8%, and 56% of Cd, Cu, Zn, Cr, Hg, and Ni are at low pollution level, respectively; 4%, 4%, and 28% of samples are moderately polluted by Cd, Hg, and Ni, respectively; 4% and 4% of samples are moderately to seriously polluted by Cd and Hg. The Ni at 4% of the sampling points exhibited heavy pollution, suggesting that there could be human activities overlapping with the geological accumulation processes in this sample point.

The $E_i$ of the eight heavy metals in oats soils were as follows: Cd (14.34) > As (3.18) > Ni (2.48) > Cu (1.45) > Cr (0.66) > Pb (0.49) > Hg (0.33) > Zn (0.24) (Figure 5b). The mean values of $E_i$ for each sampling point were less than 40, indicating the level of ecological risk was at the light degree. Cd reaches a degree of severe pollution in 4% of sampling sites; 4% of Hg values were at the medium pollution level. *RI* results revealed that 96% of sampling sites exhibited light potential ecological risk, and 4% of sampling sites were at the level of medium ecological risk. Mean contribution rates of the $E_i$ to the *RI* values of eight heavy metals were as follows: Cd (59%) > As (13%) > Ni (10%) > Cu (7%) > Hg (5%) > Cr (3%) > Pb (2%) > Zn (1%).

Health Risk Assessment

The results of health risk evaluation are presented in Table 2. From the perspective of different exposure routes, the order of HQ for the eight heavy metals for different exposure

routes of adults could be the same. For adults and children, HQ of Cu, Pb, Zn, Cr, As, Hg, and Ni decrease in the order of $HQ_{oral} > HQ_{dermal} > HQ_{inh}$. HQ values of Cd decrease in the order of $HQ_{oral} > HQ_{inh} > HQ_{dermal}$. From the perspective of different exposed populations, HQ values of eight heavy metals of children in three manners should be consider greater than adults.

**Table 2.** Non-carcinogenic risk index and carcinogenic risk index in soil planted with hulless oats.

| | **Non-Carcinogenic Risk Index** | | | | | | | |
|---|---|---|---|---|---|---|---|---|
| element | $HQ_{oral}$ | | $HQ_{inh}$ | | $HQ_{dermal}$ | | HI | |
| | adult | children | adult | children | adult | children | adult | children |
| Cd | $3.56 \times \times 10^{-10}$ | $2.54 \times 10^{-3}$ | $3.33 \times 10^{-14}$ | $5.95 \times 10^{-8}$ | $1.24 \times 10^{-14}$ | $1.63 \times 10^{-14}$ | | |
| Cu | $1.52 \times 10^{-7}$ | $1.03 \times 10^{-6}$ | $1.41 \times 10^{-11}$ | $2.52 \times 10^{-11}$ | $4.39 \times 10^{-9}$ | $5.77 \times 10^{-9}$ | | |
| Pb | $7.20 \times 10^{-9}$ | $5.14 \times 10^{-8}$ | $6.71 \times 10^{-13}$ | $1.20 \times 10^{-12}$ | $4.18 \times 10^{-12}$ | $5.50 \times 10^{-12}$ | | |
| Zn | $4.64 \times 10^{-6}$ | $3.31 \times 10^{-5}$ | $4.35 \times 10^{-10}$ | $7.76 \times 10^{-10}$ | $2.02 \times 10^{-9}$ | $2.65 \times 10^{-9}$ | $4.86 \times 10^{-6}$ | $2.58 \times 10^{-3}$ |
| Cr | $5.17 \times 10^{-8}$ | $3.69 \times 10^{-7}$ | $5.09 \times 10^{-14}$ | $9.08 \times 10^{-14}$ | $2.25 \times 10^{-12}$ | $2.96 \times 10^{-12}$ | | |
| As | $4.91 \times 10^{-10}$ | $3.51 \times 10^{-9}$ | $1.12 \times 10^{-13}$ | $2.01 \times 10^{-13}$ | $1.04 \times 10^{-11}$ | $1.37 \times 10^{-11}$ | | |
| Hg | $4.14 \times 10^{-12}$ | $2.96 \times 10^{-11}$ | $3.88 \times 10^{-16}$ | $6.94 \times 10^{-16}$ | $5.88 \times 10^{-13}$ | $7.72 \times 10^{-13}$ | | |
| Ni | $4.34 \times 10^{-9}$ | $3.03 \times 10^{-8}$ | $9.69 \times 10^{-13}$ | $1.73 \times 10^{-12}$ | $8.99 \times 10^{-11}$ | $1.18 \times 10^{-10}$ | | |
| | **Carcinogenic risk index** | | | | | | | |
| elements | $CR_{oral}$ | | $CR_{inh}$ | | $CR_{dermal}$ | | TCR | |
| | adult | children | adult | children | adult | children | adult | children |
| Cd | $9.30 \times 10^{-7}$ | $1.33 \times 10^{-6}$ | $2.57 \times 10^{-14}$ | $9.19 \times 10^{-15}$ | $8.09 \times 10^{-9}$ | $2.13 \times 10^{-9}$ | | |
| Cr | ND | ND | $2.62 \times 10^{-7}$ | $9.35 \times 10^{-8}$ | ND | ND | $1.08 \times 10^{-5}$ | $1.50 \times 10^{-5}$ |
| As | $9.47 \times 10^{-6}$ | $1.35 \times 10^{-5}$ | $2.54 \times 10^{-12}$ | $9.09 \times 10^{-13}$ | $8.24 \times 10^{-8}$ | $2.16 \times 10^{-8}$ | | |
| Ni | ND | ND | $4.29 \times 10^{-9}$ | $1.53 \times 10^{-9}$ | ND | ND | | |

Compared to adults ($4.66 \times 10^{-6}$), children ($2.47 \times 10^{-3}$) may be more sensitive to non-carcinogenic risks. However, HI among children and adults were below 1, illustrating that heavy metals in this area hardly pose non-carcinogenic risks to human health. The order of HI for the eight heavy metals in adults decrease as follows: Zn > Cu > Cr > Pb > Ni > As > Cd > Hg. The contribution rate of Zn to the HI of adults was 95.5%. Elevated ingestion of Zn will lead to fatigue, epigastric pain, vomiting, and nausea [1]. The HI of eight heavy metals in children descend in the order of Cd > Zn > Cu > Cr > Pb > Ni > As > Hg. Cd contributed 98.66% to the HI of children. Pediatric cancer and stunted development in children are known to be related to Cd exposure [56].

From the perspective of different exposure routes, the order of CR values of eight heavy metals for different exposure routes of adults could be the same. For adults and children, CR values of Cd and As were decreased in the order of $CR_{oral} > CR_{dermal} > CR_{inh}$. From the perspective of different exposed populations, the CR of Cd, Cr, As, and Ni showed differences. In the route of ingestion, the CR values for Cd and As of adults should be considered lower than in children; in the route of inhalation, the CR values about Cd, Cr, and As of adults were higher than children, which was similar to Cd and As in the route of dermal contact.

TCR results revealed that the mean TCR values of children and adults were $1.44 \times 10^{-5}$ and $1.04 \times 10^{-5}$, respectively, indicating children should be considered more sensitive to carcinogenic risks than adults. The indexes were lower than the $10^{-4}$ but higher than $10^{-6}$, indicating non-negligible carcinogenic risks in the survey region. According to the level of carcinogenic risk, the order of different metals for adults and children was As > Cd > Cr > Ni. As was the main contributing factor, accounting for 88.8% and 90.5% for adults and children, respectively. Some studies suggested that As has a strong cancer risk to humans, and exposure to As could lead to skin, lung, liver or kidney cancers [4,43]. We notice here that a high level of As was observed in most of the farmland in China [57].

*3.5. Source Identification of Heavy Metals in Oats Soil*

The major soil pollutants include Cd, Hg, and Ni elements in oats soils. The results of PCA (Table S5) suggested there were main sources of heavy metals in oats soil. PC1 was mainly comprised by Cu, Ni, Zn, and Cr, accounting for 48.9% of the total variance.

Ni pollution is generally fundamentally related to soil parent material [58]. However, Ni concentrations of all samples exceed the background values of Hebei province, indicating that Hg pollution is caused by human activities. Xiong [59] indicated that Ni is mainly caused by mining. Zhangbei is rich in zinc and copper minerals which possibly produce Zn and Cu pollution. Ru et al. [60] indicated that livestock manure including Cu, Zn, and Cr is one of the important sources of heavy metals in soil of Hebei. PC2 had the largest loadings of Pb and As. Pb is produced by gasoline combustion and battery production. The developing of wind power equipment manufacturing industrial chain and frequent use of agricultural vehicles is likely to form Pb pollution; especially, the largest values of Pb are distributed near highways. As mainly originated from livestock production [61], PC3 was dominated by Cd and Hg. Cd pollution is mainly affected by combustion of fossil fuels and inappropriate measures of agriculture [62]. Some studies showed that Cd enters into soils by the fertilizer application because it occurs as an impurity of phosphate fertilizers [63,64]. Hg pollution is ascribed to the usage of Hg-containing pesticides [65,66]. In China, atmospheric deposition is the main source of heavy metals in agricultural soils [67–69]. Zhangbei county has a long heating cycle every year. The combustion of a mass of fossil fuels and the intensification of air pollution is possible to cause accumulation of soil pollutants. The pollutants are hard to diffuse because of the Bashang plateau in the southeast of the study area. Irrigation water can be a source of pollution [70,71]. As a main source of irrigation, the river in Zhangbei county, China, is likely to contain heavy metals [61].

### 3.6. Comparison of Risk Assessment of Heavy Metals in Potato and Oats Soils

*NIPI*, geo-accumulation, and ecological risk method were applied to assess the heavy metal pollution of the two plant root soils. In the potato soils, one point was at the moderate-serious pollution according to the results of $I_{ego}$. The results of *NIPI* showed that point sources pollution of Cd, Hg, and Ni occurred in oats soils. The $I_{ego}$ evaluation indicated that one point was moderately polluted by Cd. The assessment results of *RI* showed that one point was severely polluted by Cd, and 4% of samples were moderately polluted by Hg.

Different intensity of human activities or differences in the capacity of crops to accumulate heavy metals may lead to a different status of pollution of heavy metals. The spatial distribution of heavy metals in potato and oats soils showed that high contents of heavy metals were distributed in the region with high human intervention such as population concentration or around roads. Compared to As, Cr, Hg, and Pb, Cd were to be accumulated in potato [44]. Chen et al. [72] indicated that potato tend to efficiently adsorb Cd from soils. This may lead to lower Cd concentrations in potato soils than those in oats soils. Previous studies observed that roots of oats had the potential to absorb Hg from soils [73]. Xie et al. [44] found that oats were easier to concentrate Hg, followed by Cd and Cr. It is necessary to grow the non-crop hyperaccumulator species in pollution areas for purpose of heavy metals remediation [74]. The concentrations of heavy metals in potato and oats need to be further analyzed.

### 4. Conclusions

The content and spatial distribution pattern, and ecological risk and health risk of heavy metals in potato and hulless oats soils from a cattle-producing area near the Zhangbei Bashang farmland Hebei province, China, were slightly different. The mean contents of heavy metals in the soil of both crops was less than the risk screening value, indicating that the heavy metal pollution in the study area was at a safe level as a whole. However, the values of Cd, Hg, and Ni in the oats soil in some areas exceeded the screening value.

Comprehensive ecological risk assessment by different methods. There was no obvious ecological risk in potato soil. In the oats soil, the single factor index method showed that 4%, 4%, and 8% of samples were slightly polluted by Cd, Hg, and Ni, respectively, and 4% of points were moderately contaminated by Cd; the geo-accumulation index suggests that 4% of samples is moderately polluted by Cd; according to results of $E_i$, 4% of samples were

strongly polluted by Cd and 4% of sites were moderately polluted by Hg. The oats soil in the survey region was mostly characterized by slight ecological risk.

The health risks of oats soil were evaluated. Findings indicated that the risk of non-carcinogenic and carcinogenic pathologies caused by heavy metals did not exist for adults and children; children should be considered to be in more danger with respect to both non-carcinogenic and carcinogenic risk than adults. Zn and Cd were the main non-carcinogenic pollutants for adults and children, respectively; As was the main carcinogenic pollutant. The results of PCA indicated that mining and fertilizer application could be the main pollution sources.

**Supplementary Materials:** The following supporting information can be downloaded at: https://www.mdpi.com/article/10.3390/app122312425/s1, Table S1: Risk screening values for soil contamination of agriculture land; Table S2: Classification standard of Potential Ecological Risk of heavy metal in soils; Table S3: Exposure parameter values of health risk assessment; Table S4: E Heavy metal exposure reference dose ($RFE_i$) and slope factor ($CSF_i$); Table S5: PCA analysis in oats soil.

**Author Contributions:** Writing—original draft preparation, Y.L.; investigation, Q.Z.; writing—review and editing, C.W.; conceptualization, C.W. and X.T.; validation, S.Z. All authors have read and agreed to the published version of the manuscript.

**Funding:** This research received no external funding.

**Institutional Review Board Statement:** Not applicable.

**Informed Consent Statement:** Not applicable.

**Data Availability Statement:** Not applicable.

**Conflicts of Interest:** The authors declare no conflict of interest.

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
