# Peer review of "Ecological and Health Risk Assessment of Heavy Metals in Farmland in the South of Zhangbei County, Hebei Province, China"

_applsci, doi:10.3390/app122312425_

Round 1
Reviewer 1 Report
Hello;
After reading your article, I just wanted to tell you that it is good, and it shows excellent results in the field. you just must correct the number of the title "figure 1" page 5 to "figure 2". good luck
Reviewer 2 Report
The research work and written of the manuscript is good

Reviewer 3 Report
The paper reports the content of eight heavy metals in agricultural soil of Zhangbei County, China, and uses the results to evaluate pollution indexes and health risks. The authors should explain the motivation for the choice of the specific area, and what is the interest of this study for an international audience. There is no detail of the analytical procedures, thus the quality of data remains undefined. The results are of essentially descriptive nature, with very cursory consideration of sources, processes... The study may have some interest at the local scale, but implications at a larger scale are not apparent. Finally, the paper is not well written. Many statements appear obscure or unjustified. Extensive editing by a native speaker would be required (as an example, I made some comments to the first two pages of the ms - see the attached pdf). More comments in the attached pdf.

Reviewer 4 Report
Comments:
1. Use “,” before “respectively” throughout the text.
2. Add a section entitled “study area description” in Material and methods and add information regarding the study area, population, coordinates, climates and also the main natural processes and human activities which may influence the cocnetatuions of metals.
3. Add “(c)” to Figure 3.
4. To improve the paper regarding metals, health risk and ecological risk, the authors can read and use the following papers:
-Determination of Concentration of Metals in Grapes Grown in Gonabad Vineyards and Assessment of Associated Health Risks
II- - Distribution, exposure, and human health risk analysis of heavy metals in drinking groundwater of Ghayen County, Iran
III. -Spatial distribution and contamination of heavy metals in surface water, groundwater and topsoil surrounding Moghan’s tannery site in Ardabil, Iran
IV. - Characteristics, water quality index and human health risk from nitrate and fluoride in Kakhk city and its rural areas, Iran
Round 2
Reviewer 3 Report
The paper remains very difficult to read because of the poor language, and this fact does not favor appreciation of the content. I made several language comments in the enclosed file, but there is much more to be corrected. This said, a number of important questions remain open, and suggest that the paper is not acceptable for publication in its present form.
Introduction (especially final paragraph). Unclear focus. The area is seat of both animal husbandry and crop farming. How are the two activities related? Which is the focus of the study? Apparently it concerns only some vegetables. Where is the scientific progress? I appreciate that the area is an important agricultural producer, but the study appears only preliminary (relatively small number of samples - 28 over an area of several tens of square kilometers - less than one sample per square kilometer) and limited to soil. In the discussion, the possibility is hinted of transfer to the biosphere, but no actual data are given. Sources of pollution remain speculative.
Line 107 - explain the reason for the two different sieving sizes
Fig. 1 - you should add the explanation for the clearly uneven distribution of samples
Geo-accumulation index. I must say I am a bit skeptical about the Muller's assumption that an enrichment factor of 1.5 is an indication of an anthropogenic activity - it does not take in full account natural variability. In any case, you need a local reference value to demonstrate an anomaly
Figure 2. These color maps are computational artifacts without much real significance, especially in the western section, where you are interpolating very few values spaced several kilometers apart. You just can say nothing of what there is in between, especially considering the intrinsic inhomogeneity of soil (two samples taken few meters apart may return quite different values). It would be much more meaningful to report the results of single sampling points, with colors keyed to range of metal content)
Table 1. You should consider the analytical error. You say the analyses were made in triplicate - I suppose you report here the average - which is the standard deviation?
In the health risk section, you make confusion between what is an actual finding of the paper and what is just a consequence of the assumptions in the model (e.g., the order of HQ, or the fact that the risk is deemed higher for children than for adults). Moreover, some assumptions appears as just gross approximations (e.g., exposure 365 days per year - this could be (nearly) true for people directly working in the fields, but it seems unlikely for the population at large)
The section on source identification contains many random quotations from literature that do not necessarily apply to the study area. For instance, line 353 - a geogenic origin of Ni depends VERY MUCH on the geological substrate; you say nothing about the local geology, so this statement is meaningless; line 356 - do tanneries exist in the study area?; lines 363-364: in many areas the main Hg source is atmospheric transport (e.g., from fossil fuel burning), or it could be related to specific industrial activities. In conclusion, you cannot identify any source because you have no specific data for the study area. For the same reason, the conclusion at lines 451-453 is purely speculative and not demonstrated
